# Denoising Masked AutoEncoders Help Robust Classification

**Quanlin Wu[1], Hang Ye[1], Yuntian Gu[1], Huishuai Zhang[2], Liwei Wang[3]***, **Di He[3]***
[1]Peking University  [2]Microsoft Research Asia
[3]National Key Lab of General AI, School of Artificial Intelligence, Peking University
{quanlin, yehang, dihe}@pku.edu.cn, guyuntian@stu.pku.edu.cn,
huzhang@microsoft.com, wanglw@cis.pku.edu.cn

## ABSTRACT

In this paper, we propose a new self-supervised method, which is called Denoising Masked AutoEncoders (DMAE), for learning certified robust classifiers of images. In DMAE, we corrupt each image by adding Gaussian noises to each pixel value and randomly masking several patches. A Transformer-based encoder-decoder model is then trained to reconstruct the original image from the corrupted one. In this learning paradigm, the encoder will learn to capture relevant semantics for the downstream tasks, which is also robust to Gaussian additive noises. We show that the pre-trained encoder can naturally be used as the base classifier in Gaussian smoothed models, where we can analytically compute the certified radius for any data point. Although the proposed method is simple, it yields significant performance improvement in downstream classification tasks. We show that the DMAE ViT-Base model, which just uses 1/10 parameters of the model developed in recent work (Carlini et al., 2022), achieves competitive or better certified accuracy in various settings. The DMAE ViT-Large model significantly surpasses all previous results, establishing a new state-of-the-art on ImageNet dataset. We further demonstrate that the pre-trained model has good transferability to the CIFAR-10 dataset, suggesting its wide adaptability. Models and code are available at https://github.com/quanlin-wu/dmae.

## 1 INTRODUCTION

Deep neural networks have demonstrated remarkable performance in many real applications (He et al., 2016; Devlin et al., 2019; Silver et al., 2016). However, at the same time, several works observed that the learned models are vulnerable to adversarial attacks (Szegedy et al., 2013; Biggio et al., 2013). Taking image classification as an example, given an image $x$ that is correctly classified to label $y$ by a neural network, an adversary can find a small perturbation such that the perturbed image, though visually indistinguishable from the original one, is predicted into a wrong class with high confidence by the model. Such a problem raises significant challenges in practical scenarios.

Given such a critical issue, researchers seek to learn classifiers that can provably resist adversarial attacks, which is usually referred to as certified defense. One of the seminal approaches in this direction is the Gaussian smoothed model. A Gaussian smoothed model $g$ is defined as $g(x) = \mathbb{E}_\eta f(x + \eta)$, in which $\eta \sim \mathcal{N}(0, \sigma^2 I)$ and $f$ is an arbitrary classifier, e.g., neural network. Intuitively, the smoothed classifier $g$ can be viewed as an ensemble of the predictions of $f$ that takes noise-corrupted images $x + \eta$ as inputs. Cohen et al. (2019) derived how to analytically compute the certified radius of the smoothed classifier $g$, and follow-up works improved the training methods of the Gaussian smoothed model with labeled data (Salman et al., 2019; Zhai et al., 2021; Jeong & Shin, 2020; Horváth et al., 2022; Jeong et al., 2021).

Recently, Salman et al. (2020); Carlini et al. (2022) took the first step to train Gaussian smoothed classifiers with the help of self-supervised learning. Both approaches use a compositional model architecture for $f$ and decompose the prediction process into two stages. In the first stage, a denoising

---

*Correspondence to: Di He <dihe@pku.edu.cn> and Liwei Wang <wanglw@pku.edu.cn>.

| Method | #Params | Extra data | Certified Accuracy(%) at $\ell_2$ radius $r$ | | | | |
|---|---|---|---|---|---|---|---|
| | | | 0.5 | 1.0 | 1.5 | 2.0 | 3.0 |
| RS (Cohen et al., 2019) | 26M | ✗ | 49.0 | 37.0 | 29.0 | 19.0 | 12.0 |
| SmoothAdv (Salman et al., 2019) | 26M | ✗ | 54.0 | 43.0 | 37.0 | 27.0 | 20.0 |
| Consistency (Jeong & Shin, 2020) | 26M | ✗ | 50.0 | 44.0 | 34.0 | 24.0 | 17.0 |
| MACER (Zhai et al., 2021) | 26M | ✗ | 57.0 | 43.0 | 31.0 | 25.0 | 14.0 |
| Boosting (Horváth et al., 2022) | 78M | ✗ | 57.0 | 44.6 | 38.4 | 28.6 | 20.2 |
| SmoothMix (Jeong et al., 2021) | 26M | ✗ | 50.0 | 43.0 | 38.0 | 26.0 | 17.0 |
| Diffusion+BEiT (Carlini et al., 2022) | †857M | ✓ | **71.1**[*] | 54.3 | 38.1 | 29.5 | 13.1 |
| Ours (DMAE ViT-B) | 87M | ✗ | 69.6 | **57.9**[*] | **47.8**[*] | **35.4**[*] | **22.5**[*] |
| Ours (DMAE ViT-L) | 304M | ✗ | **73.6**[**] | **64.6**[**] | **53.7**[**] | **41.5**[**] | **27.5**[**] |

Table 1: **Certified accuracy (top-1) of different models on ImageNet.** Following Carlini et al. (2022), for each noise level $\sigma$, we select the best certified accuracy from the original papers. [**] denotes the best result, and [*] denotes the second best at each $\ell_2$ radius. †Carlini et al. (2022) uses a diffusion model with 552M parameters and a BEiT-Large model with 305M parameters. It can be seen that our DMAE ViT-B/ViT-L models achieve the best performance in most of the settings.

model is used to purify the noise-corrupted inputs. Then in the second stage, a classifier is applied to predict the label from the denoised image. Since the first-stage denoising model and the second-stage classification model can be learned or benefited from standard self-supervised approaches, the smoothed classifier $g$ can achieve better performance than previous works. For example, Carlini et al. (2022) achieved 71.1% certified accuracy at $\ell_2$ radius $r = 0.5$ and 54.3% at $r = 1.0$ on ImageNet by applying a pre-trained denoising diffusion model in the first stage (Nichol & Dhariwal, 2021) and a pre-trained BEiT (Bao et al., 2021) in the second stage. Despite its impressive performance, such a two-stage process requires much more parameters and separated training.

Different from Salman et al. (2020); Carlini et al. (2022) that use two models trained for separated purposes, we believe that a single compact network (i.e., vision Transformer) has enough expressive power to learn robust feature representation with proper supervision. Motivated by the Masked AutoEncoder (MAE) (He et al., 2022), which learns latent representations by reconstructing missing pixels from masked images, we design a new self-supervised task called Denoising Masked AutoEncoder (DMAE). Given an unlabeled image, we corrupt the image by adding Gaussian noise to each pixel value and randomly masking several patches. The goal of the task is to train a model to reconstruct the clean image from the corrupted one. Similar to MAE, DMAE also intends to reconstruct the masked information; hence, it can capture relevant features of the image for downstream tasks. Furthermore, DMAE takes noisy patches as inputs and outputs denoised ones, making the learned features robust with respect to additive noises. We expect that the semantics and robustness of the representation can be learned simultaneously, enabling efficient utilization of the model parameters.

Although the proposed DMAE method is simple, it yields significant performance improvement on downstream tasks. We pre-train DMAE ViT-Base and DMAE ViT-Large, use the encoder to initialize the Gaussian smoothed classifier, and fine-tune the parameters on ImageNet. We show that the DMAE ViT-Base model with 87M parameters, one-tenth as many as the model used in Carlini et al. (2022), achieves competitive or better certified accuracy in various settings. Furthermore, the DMAE ViT-Large model (304M) significantly surpasses the state-of-the-art results in all tasks, demonstrating a single-stage model is enough to learn robust representations with proper self-supervised tasks. We also demonstrate that the pre-trained model has good transferability to other datasets. We empirically show that decent improvement can be obtained when applying it to the CIFAR-10 dataset. Model checkpoints will be released in the future.

## 2 RELATED WORK

Szegedy et al. (2013); Biggio et al. (2013) observed that standardly trained neural networks are vulnerable to adversarial attacks. Since then, many works have investigated how to improve the robustness of the trained model. One of the most successful methods is adversarial training, which adds adversarial examples to the training set to make the learned model robust against such attacks (Madry et al., 2018; Zhang et al., 2019). However, as the generation process of adversarial examples is pre-

defined during training, the learned models may be defeated by stronger attacks (Athalye et al., 2018). Therefore, it is important to develop methods that can learn models with certified robustness guarantees. Previous works provide certified guarantees by bounding the certified radius layer by layer using convex relaxation methods (Wong & Kolter, 2018; Wong et al., 2018; Weng et al., 2018; Balunovic & Vechev, 2020; Zhang et al., 2021; 2022a;b). However, such algorithms are usually computationally expensive, provide loose bounds, or have scaling issues in deep and large models.

**Randomized smoothing.** Randomized smoothing is a scalable approach to obtaining certified robustness guarantees for any neural network. The key idea of randomized smoothing is to add Gaussian noise in the input and to transform any model into a Gaussian smoothed classifier. As the Lipschitz constant of the smoothed classifier is bounded with respect to the $\ell_2$ norm, we can analytically compute a certified guarantee on small $\ell_2$ perturbations (Cohen et al., 2019). Follow-up works proposed different training strategies to maximize the certified radius, including ensemble approaches (Horváth et al., 2022), model calibrations (Jeong et al., 2021), adversarial training for smoothed models (Salman et al., 2019) and refined training objectives (Jeong & Shin, 2020; Zhai et al., 2021). Yang et al. (2020); Blum et al. (2020); Kumar et al. (2020) extended the method to general $\ell_p$ perturbations by using different shapes of noises.

**Self-supervised pre-training in vision.** Learning the representation of images from unlabeled data is an increasingly popular direction in computer vision. Mainstream approaches can be roughly categorized into two classes. One class is the contrastive learning approach which maximizes agreement between differently augmented views of an image via a contrastive loss (Chen et al., 2020; He et al., 2020). The other class is the generative learning approach, which randomly masks patches in an image and learns to generate the original one (Bao et al., 2021; He et al., 2022). Several works utilized self-supervised pre-training to improve image denoising (Joshua Batson, 2019; Yaochen Xie, 2020), and recently there have been attempts to use pre-trained denoisers to achieve certified robustness. The most relevant works are Salman et al. (2020); Carlini et al. (2022). Both works first leverage a pre-trained denoiser to purify the input, and then use a standard classifier to make predictions. We discuss these two works and ours in depth in Sec. 3.

## 3 METHOD

### 3.1 NOTATIONS AND BASICS

Denote $\boldsymbol{x} \in \mathbb{R}^d$ as the input and $y \in \mathcal{Y} = \{1, \ldots, C\}$ as the corresponding label. Denote $g : \mathbb{R}^d \to \mathcal{Y}$ as a classifier mapping $\boldsymbol{x}$ to $y$. For any $\boldsymbol{x}$, assume that an adversary can perturb $\boldsymbol{x}$ by adding an adversarial noise. The goal of the defense methods is to guarantee that the prediction $g(\boldsymbol{x})$ doesn't change when the perturbation is small. Randomized smoothing (Li et al., 2018; Cohen et al., 2019) is a technique that provides provable defenses by constructing a smoothed classifier $g$ of the form:

$$g(\boldsymbol{x}) = \arg\max_{c \in \mathcal{Y}} \mathrm{P}_{\boldsymbol{\eta}}[f(\boldsymbol{x} + \boldsymbol{\eta}) = c], \text{ where } \boldsymbol{\eta} \sim \mathcal{N}(\boldsymbol{0}, \sigma^2 \boldsymbol{I}_d). \tag{1}$$

The function $f$ is called the base classifier, which is usually parameterized by neural networks, and $\boldsymbol{\eta}$ is Gaussian noise with noise level $\sigma$. Intuitively, $g(\boldsymbol{x})$ can be considered as an ensemble classifier which returns the majority vote of $f$ when its input is sampled from a Gaussian distribution $\mathcal{N}(\boldsymbol{x}, \sigma^2 \boldsymbol{I}_d)$ centered at $\boldsymbol{x}$. Cohen et al. (2019) theoretically provided the following certified robustness guarantee for the Gaussian smoothed classifier $g$.

**Theorem 1** *(Cohen et al., 2019) Given $f$ and $g$ defined as above, assume that $g$ classifies $\boldsymbol{x}$ correctly, i.e., $\mathrm{P}_{\boldsymbol{\eta}}[f(\boldsymbol{x} + \boldsymbol{\eta}) = y] \geq \max_{y' \neq y} \mathrm{P}_{\boldsymbol{\eta}}[f(\boldsymbol{x} + \boldsymbol{\eta}) = y']$. Then for any $\boldsymbol{x}'$ satisfying $||\boldsymbol{x}' - \boldsymbol{x}||_2 \leq R$, we always have $g(\boldsymbol{x}) = g(\boldsymbol{x}')$, where*

$$R = \frac{\sigma}{2}[\Phi^{-1}(P_{\boldsymbol{\eta}}[f(\boldsymbol{x} + \boldsymbol{\eta}) = y]) - \Phi^{-1}(\max_{y' \neq y} P_{\boldsymbol{\eta}}[f(\boldsymbol{x} + \boldsymbol{\eta}) = y'])]. \tag{2}$$

$\Phi$ *is the cumulative distribution function of the standard Gaussian distribution.*

**The denoise-then-predict network structure.** Even without knowing the label, one can still evaluate the robustness of a model by checking whether it can give consistent predictions when the input is perturbed. Therefore, unlabeled data can naturally be used to improve the model's robustness (Alayrac et al., 2019; Carmon et al., 2019; Najafi et al., 2019; Zhai et al., 2019). Recently,

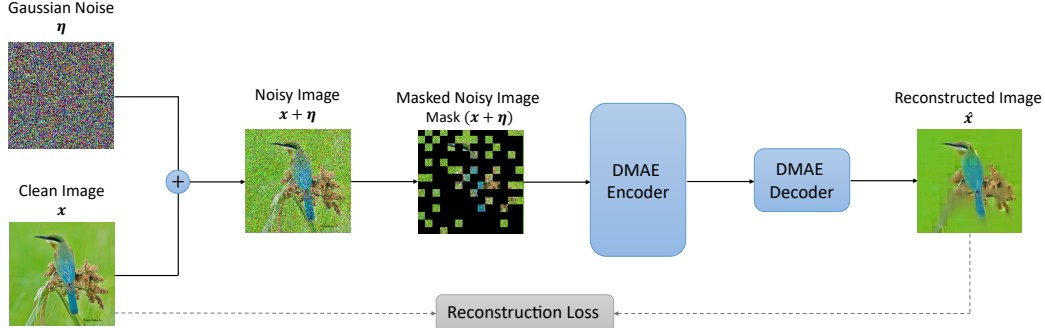

Figure 1: **Illustration of our DMAE pre-training.** We first corrupt the image by adding Gaussian noise to each pixel value, and then randomly masking several patches. The encoder and decoder are trained to reconstruct the clean image from the corrupted one.

Salman et al. (2020); Carlini et al. (2022) took steps to train Gaussian smoothed classifiers with the help of unlabeled data. Both of them use a denoise-then-predict pipeline. In detail, the base classifier $f$ consists of three components: $\theta_{\text{denoiser}}$, $\theta_{\text{encoder}}$ and $\theta_{\text{output}}$. Given any input $\boldsymbol{x}$, the classification process of $f$ is defined as below.

$$\hat{\boldsymbol{x}} = \text{Denoise}(\boldsymbol{x} + \boldsymbol{\eta}; \theta_{\text{denoiser}}) \tag{3}$$
$$\boldsymbol{h} = \text{Encode}(\hat{\boldsymbol{x}}; \theta_{\text{encoder}}) \tag{4}$$
$$y = \text{Predict}(\boldsymbol{h}; \theta_{\text{output}}) \tag{5}$$

As $f$ takes noisy image as input (see Eq.1), a denoiser with parameter $\theta_{\text{denoiser}}$ is first used to purify $\boldsymbol{x} + \boldsymbol{\eta}$ to cleaned image $\hat{\boldsymbol{x}}$. After that, $\hat{\boldsymbol{x}}$ is further encoded into contextual representation $\boldsymbol{h}$ by $\theta_{\text{encoder}}$ and the prediction can be obtained from the output head $\theta_{\text{output}}$. Note that $\theta_{\text{denoiser}}$ and $\theta_{\text{encoder}}$ can be pre-trained by self-supervised approaches. For example, one can use denoising auto-encoder (Vincent et al., 2008; 2010) or denoising diffusion model (Ho et al., 2020; Nichol & Dhariwal, 2021) to pre-train $\theta_{\text{denoiser}}$, and leverage contrastive learning (Chen et al., 2020; He et al., 2020) or masked image modelling (He et al., 2022; Xie et al., 2022) to pre-train $\theta_{\text{encoder}}$. Especially, Carlini et al. (2022) achieved state-of-the-art performance on ImageNet by applying a pre-trained denoising diffusion model as the denoiser and a pre-trained BEiT (Bao et al., 2021) as the encoder.

### 3.2 Denoising Masked Autoencoders

In the denoise-then-predict network structure above, if the denoiser is perfect, $\boldsymbol{h}$ will be robust to the Gaussian additive noise $\boldsymbol{\eta}$. Then the robust accuracy of $g$ can be as high as the standard accuracy of models trained on clean images. However, the denoiser requires a huge number of parameters to obtain acceptable results (Nichol & Dhariwal, 2021), limiting the practical usage of the compositional method in real applications.

Note that our goal is to learn representation $\boldsymbol{h}$ that captures rich semantics for classification and resists Gaussian additive noise. Using an explicit purification step before encoding is sufficient to achieve it but may not be a necessity. Instead of using multiple training stages for different purposes, we aim to adopt a single-stage approach to learn robust $\boldsymbol{h}$ through self-supervised learning directly. In particular, we extend the standard masked autoencoder with an additional denoising task, which we call the Denoising Masked AutoEncoder (DMAE). The DMAE works as follows: an image $\boldsymbol{x}$ is first divided into regular non-overlapping patches. Denote $\text{Mask}(x)$ as the operation that randomly masks patches with a pre-defined masking ratio. As shown in Fig. 1, we aim to train an autoencoder that takes $\text{Mask}(\boldsymbol{x} + \boldsymbol{\eta})$ as input and reconstructs the original image:

$$\boldsymbol{x} \rightarrow \boldsymbol{x} + \boldsymbol{\eta} \rightarrow \text{Mask}(\boldsymbol{x} + \boldsymbol{\eta}) \xrightarrow{\text{Encoder}} \boldsymbol{h} \xrightarrow{\text{Decoder}} \hat{\boldsymbol{x}}.$$

Like MAE (He et al., 2022), we adopt the asymmetric encoder-decoder design for DMAE. Both encoder and decoder use stacked Transformer layers. The encoder takes noisy unmasked patches with positional encoding as inputs and generates the representation $\boldsymbol{h}$. Then the decoder takes the representation on all patches as inputs ($\boldsymbol{h}$ for unmasked patches and a masked token embedding for

masked patches) and reconstructs the original image. Pixel-level mean square error is used as the loss function. Slightly different from MAE, the loss is calculated on all patches as the model can also learn purification on the unmasked positions. During pre-training, the encoder and decoder are jointly optimized from scratch, and the decoder will be removed while learning downstream tasks.

In order to reconstruct the original image, the encoder and the decoder have to learn semantics from the unmasked patches and remove the noise simultaneously. To enforce the encoder (but not the decoder) to learn robust semantic features, we control the capacity of the decoder by setting a smaller value of the hidden dimension and depth following He et al. (2022).

**Robust fine-tuning for downstream classification tasks.** As the encoder of DMAE already learns robust features, we can simplify the classification process of the base classifer as

$$
\boldsymbol{h} = \text{Encode}(\boldsymbol{x} + \boldsymbol{\eta}; \theta_{\text{encoder}}) \tag{6}
$$

$$
y = \text{Predict}(\boldsymbol{h}; \theta_{\text{output}}) \tag{7}
$$

To avoid any confusion, we explicitly parameterize the base classifier as $f(\boldsymbol{x}; \theta_{\text{encoder}}, \theta_{\text{output}}) = \text{Predict}(\text{Encode}(\boldsymbol{x}; \theta_{\text{encoder}}); \theta_{\text{output}})$, and denote $F(\boldsymbol{x}; \theta_{\text{encoder}}, \theta_{\text{output}})$ as the output of the last softmax layer of $f$, i.e., the probability distribution over classes. We aim to maximize the certified accuracy of the corresponding smoothed classifier $g$ by optimizing $\theta_{\text{encoder}}$ and $\theta_{\text{output}}$, where $\theta_{\text{encoder}}$ is initialized by the pre-trained DMAE model. To achieve the best performance, we use the consistency regularization training method developed in Jeong & Shin (2020) to learn $\theta_{\text{encoder}}$ and $\theta_{\text{output}}$. The loss is defined as below.

$$
\begin{aligned}
L(\boldsymbol{x}, y; \theta_{\text{encoder}}, \theta_{\text{output}}) &= \mathbb{E}_{\boldsymbol{\eta}}[\text{CrossEntropy}(F(\boldsymbol{x} + \boldsymbol{\eta}; \theta_{\text{encoder}}, \theta_{\text{output}}), y)] \\
&+ \lambda \cdot \mathbb{E}_{\boldsymbol{\eta}}[D_{\text{KL}}(\hat{F}(\boldsymbol{x}; \theta_{\text{encoder}}, \theta_{\text{output}}) \| F(\boldsymbol{x} + \boldsymbol{\eta}; \theta_{\text{encoder}}, \theta_{\text{output}}))] \\
&+ \mu \cdot H(\hat{F}(\boldsymbol{x}; \theta_{\text{encoder}}, \theta_{\text{output}}))
\end{aligned} \tag{8}
$$

where $\hat{F}(\boldsymbol{x}; \theta_{\text{encoder}}, \theta_{\text{output}}) := \mathbb{E}_{\boldsymbol{\eta} \sim \mathcal{N}(\mathbf{0}, \sigma^2 \boldsymbol{I}_d)}[F(\boldsymbol{x} + \boldsymbol{\eta}; \theta_{\text{encoder}}, \theta_{\text{output}})]$ is the average prediction distribution of the base classifier, and $\lambda, \mu > 0$ are hyperparameters. $D_{\text{KL}}(\cdot \| \cdot)$ and $H(\cdot)$ denote the Kullback–Leibler (KL) divergence and the entropy, respectively. The loss function contains three terms. Intuitively, the first term aims to maximize the accuracy of the base classifier with perturbed input. The second term attempts to regularize $F(\boldsymbol{x} + \boldsymbol{\eta}; \theta_{\text{encoder}}, \theta_{\text{output}})$ to be consistent with different $\boldsymbol{\eta}$. The last term prevents the prediction from low confidence scores. All expectations are estimated by Monte Carlo sampling.

## 4 EXPERIMENTS

In this section, we empirically evaluate our proposed DMAE on ImageNet and CIFAR-10 datasets. We also study the influence of different hyperparameters and training strategies on the final model performance. All experiments are repeated ten times with different seeds. Average performance is reported, and details can be found in the appendix.

### 4.1 PRE-TRAINING SETUP

Following He et al. (2022); Xie et al. (2022), we use ImageNet-1k as the pre-training corpus which contains 1.28 million images. All images are resized to a fixed resolution of $224 \times 224$. We utilize two vision Transformer variants as the encoder, the Base model (ViT-B) and the Large model (ViT-L) with $16 \times 16$ input patch size (Kolesnikov et al., 2021). The ViT-B encoder consists of 12 Transfomer blocks with embedding dimension 768, while the ViT-L encoder consists of 16 blocks with embedding dimension 1024. For both settings, the decoder uses 8 Transformer blocks with embedding dimension 512 and a linear projection whose number of output channels equals the number of pixel values in a patch. All the Transformer blocks have 16 attention heads. The ViT-B/ViT-L encoder have roughly 87M and 304M parameters, respectively.

For the pre-training of the two DMAE models, we set the masking ratio to 0.75 following He et al. (2022). The noise level $\sigma$ is set to 0.25. Random resizing and cropping are used as data augmentation to avoid overfitting. The ViT-B and ViT-L models are pre-trained for 1100 and 1600 epochs, where the batch size is set to 4096. We use the AdamW optimizer with $\beta_1, \beta_2 = 0.9, 0.95$, and adjust the learning rate to $1.5 \times 10^{-4}$. The weight decay factor is set to 0.05. After pre-training, we

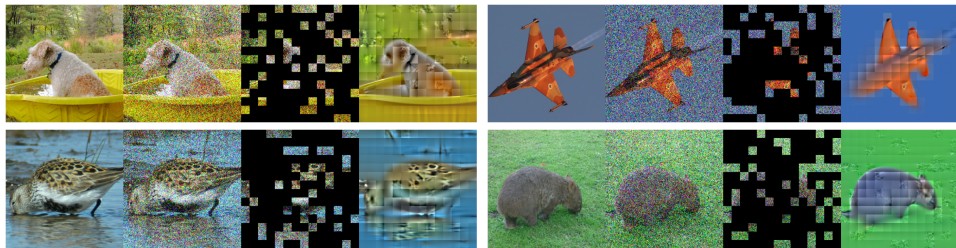

Figure 2: **Visualization.** For each group, the leftmost column shows the original image. The following two correspond to the image with Gaussian perturbation (noise level $\sigma = 0.25$) and the masked noisy image. The last column illustrates the reconstructed image by our DMAE ViT-L model.

also visualize the model performance of DMAE ViT-L in Fig. 2. From the figure, we can see that the trained model can recover the masked patches and purify the noisy unmasked patches, which demonstrates its capability of accomplishing both tasks simultaneously.

## 4.2 Fine-tuning for ImageNet classification

**Setup.** In the fine-tuning stage, we add a linear prediction head on top of the encoder for classification. The ViT-B model is fine-tuned for 100 epochs, while the ViT-L is fine-tuned for 50 epochs. Both settings use AdamW with $\beta_1, \beta_2 = 0.9, 0.999$. The weight decay factor is set to 0.05. We set the base learning rate to $5 \times 10^{-4}$ for ViT-B and $1 \times 10^{-3}$ for ViT-L. Following Bao et al. (2021), we use layer-wise learning rate decay (Kevin Clark & Manning, 2020) with an exponential rate of 0.65 for ViT-B and 0.75 for ViT-L. We apply standard augmentation for training ViT models, including RandAug (Ekin D Cubuk & Le, 2020), label smoothing (Szegedy et al., 2016), mixup (Hongyi Zhang & Lopez-Paz, 2018) and cutmix (Yun et al., 2019). Following most previous works, we conduct experiments with different noise levels $\sigma \in \{0.25, 0.5, 1.0\}$. For the consistency regularization loss terms, we set the hyperparameters $\lambda = 2.0$ and $\mu = 0.5$ for $\sigma \in \{0.25, 0.5\}$, and set $\lambda = 2.0$ and $\mu = 0.1$ for $\sigma = 1.0$.

**Evaluation.** Following previous works, we report the percentage of samples that can be certified to be robust (a.k.a certified accuracy) at radius $r$ with pre-defined values. For a fair comparison, we use the official implementation[1] of `CERTIFY` to calculate the certified radius for any data point[2], with $n = 10,000, n_0 = 100$ and $\alpha = 0.001$. The result is averaged over 1,000 images uniformly selected from ImageNet validation set, following Carlini et al. (2022).

**Results.** We list the detailed results of our model and representative baseline methods in Table 2. We also provide a summarized result that contains the best performance of different methods at each radius $r$ in Table 1. It can be seen from Table 2 that our DMAE ViT-B model significantly surpasses all baselines in all settings except Carlini et al. (2022). This clearly demonstrates the strength of self-supervised learning. Compared with Carlini et al. (2022), our model achieves better results when $r \geq 1.0$ and is slightly worse when $r$ is small. We would like to point out that the DMAE ViT-B model only uses 10% parameters compared to Carlini et al. (2022), which suggests our single-stage pre-training method is more parameter-efficient than the denoise-then-predict approach. Although the diffusion model used in Carlini et al. (2022) can be applied with different noise levels, the huge number of parameters and long inference time make it more difficult to deploy.

Our DMAE ViT-L model achieves the best performance over all prior works in all settings and boosts the certified accuracy by a significant margin when $\sigma$ and $r$ are large. For example, at $r = 1.5$, it achieves 53.7% accuracy which is 15.3% better than Boosting (Horváth et al., 2022), and it surpasses Diffusion (Carlini et al., 2022) by 12.0% at $r = 2.0$. This observation is different from the one reported in Carlini et al. (2022), where the authors found that the diffusion model coupled with an off-the-shelf BEiT only yields better performance with smaller $\sigma$ and $r$.

---

[1] https://github.com/locuslab/smoothing

[2] One may notice that randomized smoothing methods require a significant number of samples (e.g., $10^5$) for evaluation. Here the samples are used to calculate the certified radius accurately. A much smaller number of samples are enough to make predictions in practice.

| $\sigma$ | Method | Certified Accuracy(%) at $\ell_2$ radius $r$ | | | | | |
|---|---|---|---|---|---|---|---|
| | | 0.0 | 0.5 | 1.0 | 1.5 | 2.0 | 3.0 |
| 0.25 | RS (Cohen et al., 2019) | 67.0 | 49.0 | 0 | 0 | 0 | 0 |
| | SmoothAdv (Salman et al., 2019) | 63.0 | 54.0 | 0 | 0 | 0 | 0 |
| | Consistency (Jeong & Shin, 2020) | - | | | | | |
| | MACER (Zhai et al., 2021) | 68.0 | 57.0 | 0 | 0 | 0 | 0 |
| | Boosting (Horváth et al., 2022) | 65.6 | 57.0 | 0 | 0 | 0 | 0 |
| | SmoothMix (Jeong et al., 2021) | - | | | | | |
| | Diffusion+BEiT (Carlini et al., 2022) | **82.8**** | 71.1 | 0 | 0 | 0 | 0 |
| | Ours (DMAE ViT-B) | 78.1 | 69.6 | 0 | 0 | 0 | 0 |
| | Ours (DMAE ViT-L) | **81.7*** | **73.6**** | 0 | 0 | 0 | 0 |
| 0.5 | RS (Cohen et al., 2019) | 57.0 | 46.0 | 37.0 | 29.0 | 0 | 0 |
| | SmoothAdv (Salman et al., 2019) | 54.0 | 49.0 | 43.0 | 37.0 | 0 | 0 |
| | Consistency (Jeong & Shin, 2020) | 55.0 | 50.0 | 44.0 | 34.0 | 0 | 0 |
| | MACER (Zhai et al., 2021) | 64.0 | 53.0 | 43.0 | 31.0 | 0 | 0 |
| | Boosting (Horváth et al., 2022) | 57.0 | 52.0 | 44.6 | 38.4 | 0 | 0 |
| | SmoothMix (Jeong et al., 2021) | 55.0 | 50.0 | 43.0 | 38.0 | 0 | 0 |
| | Diffusion+BEiT (Carlini et al., 2022) | 77.1 | 67.8 | 54.3 | 38.1 | 0 | 0 |
| | Ours (DMAE ViT-B) | 73.0 | 64.8 | **57.9*** | 47.8 | 0 | 0 |
| | Ours (DMAE ViT-L) | 77.6 | **72.4*** | **64.6**** | **53.7**** | 0 | 0 |
| 1.0 | RS (Cohen et al., 2019) | 44.0 | 38.0 | 33.0 | 26.0 | 19.0 | 12.0 |
| | SmoothAdv (Salman et al., 2019) | 40.0 | 37.0 | 34.0 | 30.0 | 27.0 | 20.0 |
| | Consistency (Jeong & Shin, 2020) | 41.0 | 37.0 | 32.0 | 28.0 | 24.0 | 17.0 |
| | MACER (Zhai et al., 2021) | 48.0 | 43.0 | 36.0 | 30.0 | 25.0 | 14.0 |
| | Boosting (Horváth et al., 2022) | 44.6 | 40.2 | 37.2 | 34.0 | 28.6 | 20.2 |
| | SmoothMix (Jeong et al., 2021) | 40.0 | 37.0 | 34.0 | 30.0 | 26.0 | 20.0 |
| | Diffusion+BEiT (Carlini et al., 2022) | 60.0 | 50.0 | 42.0 | 35.5 | 29.5 | 13.1 |
| | Ours (DMAE ViT-B) | 58.0 | 53.3 | 47.1 | 41.7 | **35.4*** | **22.5*** |
| | Ours (DMAE ViT-L) | 65.7 | 59.0 | 53.0 | **47.9*** | **41.5**** | **27.5**** |

Table 2: **Certified accuracy (top-1) of different models on ImageNet with different noise levels.** ** denotes the best result, and * denotes the second best at each radius $r$.

### 4.3 FINE-TUNING FOR CIFAR-10 CLASSIFICATION

**Setup.** We show the DMAE models can benefit not only ImageNet but also the CIFAR-10 classification tasks, suggesting the nice transferability of our pre-trained models. We use the DMAE ViT-B checkpoint as a showcase. As the sizes of the images in ImageNet and CIFAR-10 are different, we pre-process the images CIFAR-10 to $224 \times 224$ to match the pre-trained model. Note that the data distributions of ImageNet and CIFAR-10 are far different. To address this significant distributional shift, we *continue pre-training* the DMAE model on the CIFAR-10 dataset. We set the continued pre-training stage to 50 epochs, the base learning rate to $5 \times 10^{-5}$, and the batch size to 512. Most of the fine-tuning details is the same as that on ImageNet in Sec. 4.2, except that we use a smaller batch size of 256, apply only the random horizontal flipping as data augmentation, and reduce the number of the fine-tuning epochs to 50.

**Result.** The evaluation protocol is the same as that on ImageNet in Sec. 4.2. We draw $n = 100,000$ noise samples and report results averaged over the entire CIFAR-10 test set. The results are presented in Table 3. Without continued pre-training, our DMAE ViT-B model still yields comparable performance with Carlini et al. (2022), and the model outperforms it when continued pre-training is applied. It is worth noting that the number of parameters of Carlini et al. (2022) is larger, and the diffusion model is trained on CIFAR datasets. In comparison, our model only uses a smaller amount of parameters, and the pre-trained checkpoint is directly borrowed from Sec. 4.1. Our model performance is significantly better than the original consistent regularization method (Jeong & Shin, 2020), demonstrating the transferability of the pre-training model. Specifically, our method outperforms the original consistent regularization by $12.0\%$ at $r = 0.25$, and by $9.0\%$ at $r = 0.5$. We believe our pre-trained checkpoint can also improve other baseline methods to achieve better results.

| Method | Params | Extra data | Certified Accuracy(%) at $\ell_2$ radius $r$ | | | |
|---|---|---|---|---|---|---|
| | | | 0.25 | 0.5 | 0.75 | 1.0 |
| RS (Cohen et al., 2019) | 1.7M | ✗ | 61.0 | 43.0 | 32.0 | 23.0 |
| SmoothAdv (Salman et al., 2019) | 1.7M | ✓ | 74.8 | 60.8 | 47.0 | 37.8 |
| Consistency (Jeong & Shin, 2020) | 1.7M | ✗ | 68.8 | 58.1 | 48.5 | 37.8 |
| MACER (Zhai et al., 2021) | 1.7M | ✗ | 71.0 | 59.0 | 46.0 | **38.0**$^*$ |
| Boosting (Horváth et al., 2022) | 17M | ✗ | 70.6 | 60.4 | **52.4**$^{**}$ | **38.8**$^{**}$ |
| SmoothMix (Jeong et al., 2021) | 1.7M | ✗ | 67.9 | 57.9 | 47.7 | 37.2 |
| Diffusion (Carlini et al., 2022) | †137M | ✓ | **79.3**$^*$ | **65.5**$^*$ | 48.7 | 35.5 |
| Ours (DMAE ViT-B) | 87M | ✓ | 79.2 | 64.6 | 47.3 | 36.1 |
| +continued pre-training | 87M | ✓ | **80.8**$^{**}$ | **67.1**$^{**}$ | **49.7**$^*$ | 37.7 |

Table 3: **Certified accuracy (top-1) of different models on CIFAR-10.** Each entry lists the certified accuracy of best Gaussian noise level $\sigma$ from the original papers. $^{**}$ denotes the best result and $^*$ denotes the second best at each $\ell_2$ radius. †(Carlini et al., 2022) uses a 50M-parameter diffusion model and a 87M-parameter ViT-B model.

| $\sigma$ | Method | Certified Accuracy(%) at $\ell_2$ radius $r$ | | | | | |
|---|---|---|---|---|---|---|---|
| | | 0.0 | 0.5 | 1.0 | 1.5 | 2.0 | 3.0 |
| 0.25 | MAE | 32.5 | 13.3 | 0 | 0 | 0 | 0 |
| | DMAE | $58.7_{(+26.2)}$ | $45.3_{(+32.0)}$ | 0 | 0 | 0 | 0 |
| 0.5 | MAE | 13.3 | 6.9 | 2.6 | 0.1 | 0 | 0 |
| | DMAE | $26.4_{(+13.1)}$ | $16.7_{(+9.8)}$ | $10.6_{(+8.0)}$ | $5.1_{(+5.0)}$ | 0 | 0 |
| 1.0 | MAE | 3.0 | 2.2 | 1.2 | 0.5 | 0.4 | 0 |
| | DMAE | $7.6_{(+4.6)}$ | $5.4_{(+3.2)}$ | $3.5_{(+2.3)}$ | $2.4_{(+1.9)}$ | $1.6_{(+1.2)}$ | $0.4_{(+0.4)}$ |

Table 4: **DMAE v.s. MAE by linear probing on ImageNet.** Our proposed DMAE is significantly better than MAE on the ImageNet classification task, indicating that the proposed pre-training method is effective and learns more robust features. Numbers in (.) is the gap between the two methods in the same setting.

## 4.4 DISCUSSION

In this section, we discuss several design choices in our methods.

**Whether DMAE learns more robust features than MAE.** Compared with MAE, we additionally use a denoising objective in pre-training to learn robust features. Therefore, we need to examine the quality of the representation learned by DMAE and MAE to investigate whether the proposed objective helps. For a fair comparison, we compare our DMAE ViT-B model with the MAE ViT-B checkpoint released by He et al. (2022) in the linear probing setting on ImageNet. Linear probing is a popular scheme to compare the representation learned by different models, where we freeze the parameters of the pre-trained encoders and use a linear layer with batch normalization to make predictions. For both DMAE and MAE, we train the linear layer for 90 epochs with a base learning rate of 0.1. The weight decay factor is set to 0.0. As overfitting seldom occurs in linear probing, we only apply random resizing and cropping as data augmentation and use a large batch size of 16,384.

As shown in Table 4, our DMAE outperforms MAE by a large margin in linear probing. For example, with Gaussian noise magnitude $\sigma = 0.25$, DMAE can achieve $45.3\%$ certified accuracy at $r = 0.5$, 32.0 points higher than that of MAE. Note that even our models were pre-trained with a small magnitude of Gaussian noise ($\sigma = 0.25$), they still yield much better results than that of MAE under large Gaussian noise ($\sigma = 0.5, 1.0$). This clearly indicates that our method learns much more robust features compared with MAE.

**Effects of the pre-training steps.** Many previous works observe that longer pre-training steps usually helps the model perform better on downstream tasks. To investigate whether this phenomenon

| $\sigma$ | Epochs | Certified Accuracy(%) at $\ell_2$ radius $r$ | | | | | |
|---|---|---|---|---|---|---|---|
| | | 0.0 | 0.5 | 1.0 | 1.5 | 2.0 | 3.0 |
| 0.25 | 700 | 78.8 | 68.8 | 0 | 0 | 0 | 0 |
| | 1100 | $78.1_{(-0.7)}$ | $69.6_{(+0.8)}$ | 0 | 0 | 0 | 0 |
| 0.5 | 700 | 72.0 | 64.4 | 55.5 | 45.3 | 0 | 0 |
| | 1100 | $73.0_{(+1.0)}$ | $64.8_{(+0.4)}$ | $57.9_{(+2.4)}$ | $47.8_{(+2.5)}$ | 0 | 0 |
| 1.0 | 700 | 56.4 | 50.7 | 46.2 | 40.4 | 34.9 | 23.3 |
| | 1100 | $58.0_{(+1.6)}$ | $53.3_{(+2.6)}$ | $47.1_{(+0.9)}$ | $41.7_{(+1.3)}$ | $35.4_{(+0.5)}$ | $22.5_{(-0.8)}$ |

Table 5: **Effects of the pre-training steps.** From the table, we can see that the 1100-epoch model consistently outperforms the 700-epoch model in almost all settings, demonstrating that longer pre-training leads to better downstream task performance. Numbers in (.) is the gap between the two methods in the same setting.

| $\sigma$ | Method | Certified Accuracy(%) at $\ell_2$ radius $r$ | | | | | |
|---|---|---|---|---|---|---|---|
| | | 0.0 | 0.5 | 1.0 | 1.5 | 2.0 | 3.0 |
| 0.25 | DMAE-L+RS | 81.8 | 72.6 | 0 | 0 | 0 | 0 |
| | DMAE-L+CR | $81.7_{(-0.1)}$ | $73.6_{(+1.0)}$ | 0 | 0 | 0 | 0 |
| 0.5 | DMAE-L+RS | 76.4 | 69.5 | 59.3 | 45.8 | 0 | 0 |
| | DMAE-L+CR | $77.6_{(+1.2)}$ | $72.4_{(+2.9)}$ | $64.6_{(+5.3)}$ | $53.7_{(+7.9)}$ | 0 | 0 |
| 1.0 | DMAE-L+RS | 63.2 | 57.0 | 49.8 | 42.9 | 35.9 | 21.8 |
| | DMAE-L+CR | $65.7_{(+2.5)}$ | $59.0_{(+2.0)}$ | $53.0_{(+3.2)}$ | $47.9_{(+5.0)}$ | $41.5_{(+5.6)}$ | $27.5_{(+5.7)}$ |

Table 6: **DMAE with different fine-tuning methods.** From the table, we can see that our pre-trained model is compatible with different fine-tuning methods. Numbers in (.) is the gap between the two methods in the same setting.

happens in our setting, we also conduct experiments to study the downstream performance of model checkpoints at different pre-training steps. In particular, we compare the DMAE ViT-B model (1100 epochs) trained in Sec. 4.1 with an early checkpoint (700 epochs). Both models are fine-tuned under the same configuration. All results on ImageNet are presented in Table 5. It shows that the 1100-epoch model consistently outperforms its 700-epoch counterpart in almost all settings.

**Other fine-tuning methods.** In the main experiment, we use Consistency Regularization (CR) in the fine-tuning stage, and one may be interested in how much the pre-trained model can improve with other methods. To study this, we fine-tune our pre-trained DMAE ViT-L model with the RS algorithm (Cohen et al., 2019), where the only loss used in training is the standard cross-entropy classification loss in Eq.7. For this experiment, we use the same configuration as in Sec. 4.2. The results are provided in Table 6. First, we can see that the regularization loss consistently leads to better certified accuracy. In particular, it yields up to 3-5% improvement at a larger $\ell_2$ radius (r $\geq 1.0$). Second, it can also be seen that the RS model fine-tuned on DMAE ViT-L significantly surpasses lots of baselines on ImageNet. This suggests that our pre-trained DMAE ViT-L model may be combined with other training methods in the literature to improve their performance.

## 5 CONCLUSION

This paper proposes a new self-supervised method, Denoising Masked AutoEncoders (DMAE), for learning certified robust classifiers of images. DMAE corrupts each image by adding Gaussian noises to each pixel value and randomly masking several patches. A vision Transformer is then trained to reconstruct the original image from the corrupted one. The pre-trained encoder of DMAE can naturally be used as the base classifier in Gaussian smoothed models to achieve certified robustness. Extensive experiments show that the pre-trained model is parameter-efficient, achieves state-of-the-art performance, and has nice transferability. We believe that the pre-trained model has great potential in many aspects. We plan to apply the pre-trained model to more tasks, including image segmentation and detection, and investigate the interpretability of the models in the future.

## ACKNOWLEDGEMENT

This work is partially supported by the National Key R&D Program of China (2022ZD0160304). The work is supported by National Science Foundation of China (NSFC62276005), The Major Key Project of PCL (PCL2021A12), Exploratory Research Project of Zhejiang Lab (No. 2022RC0AN02), and Project 2020BD006 supported by PKUBaidu Fund. We thank all the anonymous reviewers for the very careful and detailed reviews as well as the valuable suggestions. Their help has further enhanced our work.

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

## A   APPENDIX

We present the full settings of pre-training and fine-tuning in Table 7 and Table 8, respectively.

| config | value |
|---|---|
| optimizer | AdamW |
| base learning rate | 1.5e-4 |
| weight decay | 0.05 |
| optimizer momentum | $\beta_1, \beta_2 = 0.9, 0.95$ |
| batch size | 4096 |
| learning rate schedule | cosine decay |
| warmup epochs | 40 |
| augmentation | RandomResizedCrop |
| gaussian noise | $\sigma = 0.25$ |

Table 7: Robust pre-train setting.

| config | value |
|---|---|
| optimizer | AdamW |
| base learning rate | 5e-4 (B), 1e-3 (L) |
| weight decay | 0.05 |
| optimizer momentum | $\beta_1, \beta_2 = 0.9, 0.999$ |
| layer-wise lr decay | 0.65 (B), 0.75 (L) |
| batch size | 1024 |
| learning rate schedule | cosine decay |
| warmup epochs | 5 |
| training epochs | 100 (B), 50 (L) |
| augmentation | RandAug (9, 0.5) |
| label smoothing | 0.1 |
| mixup | 0.8 |
| cutmix | 1.0 |
| drop path | 0.1 (B), 0.2 (L) |
| gaussian noise | $\sigma \in \{0.25, 0.5, 1.0\}$ |
| consistency regularization | $\lambda = 2, \eta = 0.5$ ($\sigma \in \{0.25, 0.5\}$) $\lambda = 2, \eta = 0.1$ ($\sigma = 1.0$) |

Table 8: Fine-tuning setting.

## B   EVALUATION PROTOCOL

In this section, we describe the details of how to estimate the approximate certified radius and certified accuracy.

Recall that in Theorem 1, the certified radius $R$ for datapoint $\boldsymbol{x}$ can be expressed as,

$$R = \frac{\sigma}{2}[\Phi^{-1}(P_{\boldsymbol{\eta}}[f(\boldsymbol{x} + \boldsymbol{\eta}) = y]) - \Phi^{-1}(\max_{y' \neq y} P_{\boldsymbol{\eta}}[f(\boldsymbol{x} + \boldsymbol{\eta}) = y'])].$$

For computational convenience, one can obtain a bit loose, but simpler bound of certified radius by using a upper bound $1 - P_{\boldsymbol{\eta}}[f(\boldsymbol{x} + \boldsymbol{\eta}) = y] \geq \max_{y' \neq y} P_{\boldsymbol{\eta}}[f(\boldsymbol{x} + \boldsymbol{\eta}) = y']$,

$$R \geq \frac{\sigma}{2}[\Phi^{-1}(P_{\boldsymbol{\eta}}[f(\boldsymbol{x}+\boldsymbol{\eta}) = y]) - \Phi^{-1}(1 - P_{\boldsymbol{\eta}}[f(\boldsymbol{x}+\boldsymbol{\eta}) = y])] = \sigma \cdot \Phi^{-1}(P_{\boldsymbol{\eta}}[f(\boldsymbol{x}+\boldsymbol{\eta}) = y]) =: \underline{R}.$$

| $\sigma$ | Method | Certified Accuracy(%) at $\ell_2$ radius $r$ | | | | | |
|---|---|---|---|---|---|---|---|
| | | 0.0 | 0.5 | 1.0 | 1.5 | 2.0 | 3.0 |
| 0.5 | MAE-B+RS | 72.2 | 61.8 | 51.5 | 37.7 | 0 | 0 |
| | DMAE-B+RS | $72.6_{(+0.4)}$ | $63.0_{(+1.2)}$ | $53.2_{(+1.7)}$ | $39.7_{(+2.0)}$ | 0 | 0 |
| | MAE-B+CR | 72.0 | 64.1 | 55.9 | 46.2 | 0 | 0 |
| | DMAE-B+CR | $73.0_{(+1.0)}$ | $64.8_{(+0.7)}$ | $57.9_{(+2.0)}$ | $47.8_{(+1.6)}$ | 0 | 0 |

Table 9: **DMAE v.s. MAE by Fine-tuning on ImageNet.** Our proposed DMAE is consistently better than MAE on the ImageNet classification task, indicating that the proposed pre-training method is effective and learns more robust features. Numbers in (.) is the gap between the two methods in the same setting.

| Finetune $\sigma$ | Evaluate $\sigma$ | Certified Accuracy(%) at $\ell_2$ radius $r$ | | | | |
|---|---|---|---|---|---|---|
| | | 0.0 | 0.25 | 0.5 | 0.75 | 1.0 |
| 0.25 | 0.25 | **90.4** | **81.2** | 65.3 | 44.2 | 0 |
| | 0.5 | 44.5 | 35.4 | 26.8 | 18.8 | 11.3 |
| 0.5 | 0.25 | 72.2 | 65.2 | 54.1 | 41.6 | 0 |
| | 0.5 | 74.5 | 66.5 | 57.3 | **46.3** | 34.7 |
| [0, 0.75] | 0.25 | 90.2 | 80.9 | **66.1** | 45.2 | 0 |
| | 0.5 | 75.6 | 67.4 | 58.5 | 45.7 | **35.5** |

Table 10: **Certified accuracy of DMAE for different evaluation perturbations on CIFAR-10.** [0, 0.75] means training our model with $\sigma$ uniformly selectly from [0, 0.75]. The results show one model is sufficient to tackle all evaluation settings.

We use the official implementation[3] of `CERTIFY` to calculate the certified radius for any data point. For one data point $\boldsymbol{x}$, the smoothed classifier samples $n_0 = 100$ points from the noisy Gaussian distribution $\mathcal{N}(\boldsymbol{x}, \sigma^2 \boldsymbol{I}_d)$ and then votes the predicted class, while we draw $n = 10,000$ samples (for ImageNet) to estimate the lower bound of $P_{\boldsymbol{\eta}}[f(\boldsymbol{x} + \boldsymbol{\eta}) = y]$ and certify the robustness. Following previous works, we report the percentage of samples that can be certified to be robust (a.k.a certified accuracy) at radius $r$ with pre-defined values.

## C   SUPPLEMENTARY EXPERIMENTS

In this section, we report the results of several supplementary experiments.

**More comparison with MAE.** For a fair comparison, we compare our DMAE ViT-B with MAE ViT-B in the fine-tuning setting on ImageNet, in addition to the linear probing. The checkpoints are fine-tuned by the RS and CR method described in 4.4. In Table 9, we can see that DMAE outperforms MAE on all radii, which indicates the effectiveness of the denoising pre-training task.

**Fine-tuning with various levels of noise.** In the previous experiments, all the models are fine-tuned and evaluated under a specific level of Gaussian noise. One may wonder whether a single model can perform well under various levels of noise. To investigate this, we've conducted a tiny experiment on the CIFAR-10 dataset (we draw fewer noise samples $n = 10,000$ and report results averaged over 1000 images) and reported the results in Table 10. Specifically, we sample the noise scale $\sigma$ from a uniform distribution over an interval ($\sigma \in [0, 0.75]$) during fine-tuning, resulting in a robust classifier under different magnitudes of adversarial perturbations. And we compute the certified radius with this single model. The evaluation results show that it even outperforms the original model trained with a fixed noise scale when $r = 1.0$, which suggests that we can indeed use one DMAE across different settings without retraining.

---

[3] https://github.com/locuslab/smoothing

