# OpenReview forum: "Denoising Masked Autoencoders Help Robust Classification"
_ICLR.cc/2023/Conference — ICLR 2023 poster_

### Official Review · Reviewer_cA4Y · 2022-10-23

**Confidence:** 4
**Correctness:** 4
**Technical Novelty And Significance:** 2
**Empirical Novelty And Significance:** 3
**Recommendation:** 6

**Clarity, Quality, Novelty And Reproducibility:**

This work presents their methods very clearly. The detailed training process and parameters are listed, so it should not be hard to reproduce. Evaluation of novelty has been given in 'Strength And Weaknesses'.

**Strength And Weaknesses:**

Strengths:
1. Writing is easy to follow.
2. The method is simple yet effective.
3. Experiments and ablations are solid to support their arguments.

Weaknesses:
1. The method is highly established on MAE and previously proposed CR loss. The noise in pertaining can also be viewed as Gaussian noise data augmentation. There are not so many novel points.

**Summary Of The Paper:**

This paper proposes a variant of masked autoencoders (MAE), which is specifically designed for certified robustness tasks. Specifically, they add Gaussian noise in the pretaining process and use the consistency regularization method for finetuning. With a much smaller computational complexity, they obtain better results than the previous best results obtained by diffusion models.

**Summary Of The Review:**

This paper proposes a simple yet effective self-supervised pretraining framework based on MAE. Although the key techniques are not new, I still would like to accept this paper because they are the first to come up with these ideas and make them work for certified robustness on large-scale datasets.

---

> ### Author Response · Authors · 2022-11-24
> **To Reviewer cA4Y**
>
> Thank you for your appreciation of our work. We will make further explorations in this field and try to overcome the limitations.

---

### Official Review · Reviewer_E8Qx · 2022-10-23

**Confidence:** 4
**Correctness:** 3
**Technical Novelty And Significance:** 3
**Empirical Novelty And Significance:** 4
**Recommendation:** 8

**Clarity, Quality, Novelty And Reproducibility:**

The paper is mostly well written and the proposed method is clear and novel.
There is one point that could confuse people. The authors mentioned in both Intro and Method that Intro: "such a two-stage process requires much more parameters and separated training" and "Carlini et al. (2022) took steps to train Gaussian smoothed classifiers with the help of unlabeled data". This is not precise because Carlini 2022 only uses pretrained model and does not need any training for certified robustness. I understand that what the authors really want to say is that the off-the-shelf models used by Carlini 2022 need to be trained for two objectives, but the authors should make it clearer.


**Strength And Weaknesses:**

Strength: The proposed method is simple and effective.
1) It seems to be a natural fit for randomized smoothing, and I agree with the authors that there should be a more compact architecture than "denoise + predict".
2) The results on $\ell_2$ certified robustness surpasses previous works on ImageNet.

Weaknesses:
1) Explanation of why the proposed method achieves such good performance is desired.
I am not sure if there is a more principled explanation on why a more compact architecture is sufficient to learn robust classifiers than than "denoise + predict". The explanation could be from a theory or experimental perspective. For instance, in Figure 2, the visualization of the reconstructed images seems to have some artifacts (the grid pattern). If one feeds the reconstructed image to a pretrained BEiT, I suspect the classification performance is not going to be good. However, it seems that directly using the encodings that learned from the encoder (the proposed method) gives good performance. Are there more principled explanations for that?
2) More ablation studies are needed.
-  In table 2, the proposed method needs to be retrained for different noise level. However, Carlini 2022 uses one model across all different evaluation settings. It'd be good to see the certified acc of DMAE for one model trained with one noise level but evaluated on different perturbation radii.
-  Inference time is a key consideration in randomized smoothing (RS) since the large number of samples needed in RS. It'd be good to provide some quantitative evaluations for the inference time comparing with methods that use pretrained diffusion models. Comparing with standard diffusion denoising process, Carlini 2022 uses one-shot denoising instead of iterative denoising, which improves the inference speed. This paper directly uses a ViT based architecture, I am wondering if the inference time will be faster.

**Summary Of The Paper:**

This paper proposed Denoising Masked Autoencoders (DMAE). DMAE added denoising training objective to MAE, and shows superior performance against previous works on $\ell_2$ certified robustness tasks.

**Summary Of The Review:**

This paper proposed a simple and effective method to train certifiably robust classifiers. Despite the simplicity of the proposed method, it outperforms previous works. More principled explanations of why the method works and more ablation studies are desired.

---

> ### Author Response · Authors · 2022-11-16
> **To Reviewer E8Qx**
>
> We thank the reviewer for pointing out the issues. In the following, we respond to the concerns raised by the reviewer.
>
> **Q1:** Explanation of why the proposed method achieves such good performance is desired. For instance, the visualization of the reconstructed images seems to have some artifacts (the grid pattern). If one feeds the reconstructed image to a pre-trained BEiT, I suspect the classification performance is not going to be good. However, it seems that the proposed method gives a good performance.
>
> **A1:**
> To start with, it's important to note that we randomly mask the patches with a high masking ratio of $75\%$ for pre-training, whereas the entire unmasked image is fed into the encoder in the fine-tuning step. The reconstruction quality is less than ideal, as we propose a very challenging pre-training task. Actually, we can observe similar artifacts such as grid patterns in the visualization results of the original MAE, even in the absence of additive Gaussian noises.
>
> The empirical results demonstrate that our pre-trained encoder can successfully capture the robust and useful features, which suffices to yield promising classification performance. Furthermore, as we have stated in our paper, a compact architecture performs better than the denoise-then-predict pipeline because the latter suffers from accumulated errors.
>
> **Q2:** It'd be good to see the certified acc of DMAE for one model trained with one noise level but evaluated on different perturbation radii.
>
> **A2:** Thanks for the suggestion. We've conducted additional experiments on the CIFAR-10 dataset and reported the results in the revision. Specifically, we sample the noise scale $\sigma$ from a uniform distribution over an interval (e.g. $\sigma \in [0, 0.75]$]) during fine-tuning, resulting in a robust classifier under different magnitudes of adversarial perturbations. And we compute the certified radius with this single model. The evaluation results show that it even outperforms the original model trained with a fixed noise scale when $r=1.0$, which suggests that we can indeed use one DMAE across different settings without retraining.
>
>
> **Q3:** Discussion about the inference time.
>
> **A3:** The main advantage of our compact architecture over the denoise-then-predict pipeline lies in fewer parameters, which boosts the efficiency of inference. Carlini et.al. utilized the large diffusion model to purify the noisy images, leading to extra time cost in spite of one-shot denoising. On ImageNet, we obtain a throughput of 288 images per second using the ViT-L classifier on one GPU of GeForce RTX 3090 at a batch size of 32, about 13 times faster than Carlini et.al. (21 images per second at a batch size of 32 on an A100 GPU).
>
> On CIFAR-10 dataset, We obtain a throughput of 935 images per second with the ViT-B classifier on a GeForce RTX 3090 GPU at a batch size of 1,000, slightly faster than Carlini et.al. (825 images per second at a batch size of 1000 on an A100 GPU).

---

### Official Review · Reviewer_xxcH · 2022-10-25

**Confidence:** 4
**Correctness:** 2
**Technical Novelty And Significance:** 2
**Empirical Novelty And Significance:** 2
**Recommendation:** 5

**Clarity, Quality, Novelty And Reproducibility:**

Clarity: clear presentation
Quality: Applicability and comparison to denoised randomized smoothing need more justification
Novelty: mediocre
Reproducibility: code not provided; unable to verify

**Strength And Weaknesses:**

Strength: Paper is easy to follow.

Weakness: The framework does not seem to be easily extendible to any pre-trained classifier, while in general denoised randomized smoothing methods can apply to any arbitrary classifier. Therefore, the scope is limited and the comparison is not entirely fair.

**Summary Of The Paper:**

This paper proposed Denoising Masked AutoEncoders (DMAE), a vision-transformer based neural network model, and showed that the certified robustness using randomized smoothing can be either comparable or better than state-of-the-art denoised randomized smoothing methods, such as Carlini et al., 2022.

**Summary Of The Review:**

There are two major issues that I perceived and prevented me from giving a better recommendation.

1. Limited applicability: The DMAE appears to be only applicable to vision transformer-based encoder-decoder and not directly extendible to other neural network models. On the other hand, denoised randomized smoothing applies to any pre-trained classifier and can certify different architectures. My one-sentence summary of this work's major contribution would be "If one applies randomized smoothing on DMAE, it will get better or comparable certified accuracy than other (general-purpose) denoised smoothed models". However, this argument ignores the fact that denoised smoothing applies to any arbitrary classifier while the proposed model does not.

2. Unfair evaluation: Continue on my point #1, given that there is no fixed classifier in the proposed DMAE setting, I don't think comparing certified accuracy or model size makes any sense because it boils down to simply comparing the certified robustness a model (DAME) to different models (e.g., a fine-tuned vision transformer classifier). The results are not compared in a common setting and I don't see any new insights other than DAME shows good certified performance. If the goal is to show DAME is easier to certify, then more discussion on why this architecture is preferable to certification is required. But this aspect is lacking in the current manuscript. Overall, I would suggest the authors put some deep thoughts into making the comparison meaningful (e.g., trying to certify the same "classifier" while allowing the denoiser to vary), or making a deeper analysis of the source of better-certified robustness from the proposed model (again, proper baseline models/architectures would be needed here).

---

> ### Author Response · Authors · 2022-11-16
> **To Reviewer xxcH**
>
> We thank the reviewer for pointing out the issues. In the following, we respond to the concerns raised by the reviewer.
>
> **Q1:** Limited applicability: The DMAE appears to be only applicable to vision transformer-based encoder-decoder and can not directly be extended to other neural network models.
>
> **A1:** Indeed, only an encoder-decoder structure is required in our paradigm, it does not necessarily have to be a vision transformer. We choose ViT as our backbone not only because it has stronger expressive power but following the common practice in the community. (MAE, BEiT)
>
> **Q2:** Unfair evaluation: Given that there is no fixed classifier in the proposed DMAE setting, I don't think comparing certified accuracy or model size makes any sense because it boils down to simply comparing the certified robustness of a model (DAME) to different models (e.g., a fine-tuned vision transformer classifier).
>
> **A2:**
> First, it's worth noting that though we've listed several baselines which choose ResNet-50 as the base classifier, we particularly focus on the comparison between our model and Carlini et.al., which utilizes a diffusion model plus a BEiT classifier. Moreover, in Sec 4.4, we've drawn a fair comparison between DMAE and MAE in the linear probing setting. The empirical results demonstrate that the additional denoising objective in pre-training significantly improves the certified accuracy.
>
> Furthermore, we've conducted a supplemental experiment to provide more evidence. To be specific, we compare the performance of DMAE and MAE in the different fine-tuning settings (RS and CR). The results show that the proposed DMAE outperforms MAE consistently, which also supports our arguments. Note that we've added the experimental results in the appendix.

---

> > ### Comment · Reviewer_xxcH · 2022-11-18
> > **Thank you for your response**
> >
> > I thank the authors for the response and find the new experiments convincing. Though I still don't think a fair comparison can be made between the proposed encoder-decoder-based architecture versus a denoiser + a classifier as. But this aspect does not affect my rating.
> >
> > I've increased my score, but I do find that I share the same comment with multiple reviewers (Reviewer E8Qx & Reviewer Gj4g) that more explanations on why DMAE can help certified robustness are important. I am also not quite sure about the statement made by the authors that " a compact architecture performs better than the denoise-then-predict pipeline because the latter suffers from accumulated errors" in the rebuttal to Reviewer E8Qx.

---

> > > ### Author Response · Authors · 2022-11-24
> > > **To Reviewer xxcH**
> > >
> > > Thank you for the comments. We found the reviewer still has concerns about why we suggest using a compact model, and here we briefly explain the reason.
> > >
> > > First of all, it is widely believed that neural networks are usually highly over-parameterized, and there exists a "small" network that can achieve similar performance. Hinton et al.(2015) have shown that knowledge can be well transferred from an ensemble or a large network into a smaller, distilled model. And Frankle et al.(2019) stated that the well-initialized sub-networks reliably converge successfully, often faster than the original network at the same level of accuracy.
> > >
> > > Second, specifically for this problem, achieving certified robustness may not require perfect image purification. Carlini et al. utilized a huge diffusion model for image purification, with parameters as many as ***64.4\%*** (552M of 857M) of the overall model. As discussed in the paper, a robust feature representaion is the key that one needs. That's why we believe a more compact model to generate robust representation can achieve the goal achieved by robust pre-training.
> > >
> > > We hope our explanation can address the concern and you can reevaluate our work.
> > >
> > > [1] Hinton et al. Distilling the Knowledge in a Neural Network. NeurIPS 2014 Deep Learning Workshop.
> > >
> > > [2] Frankle et al. The Lottery Ticket Hypothesis: Training Pruned Neural Networks. ICLR 2019.

---

### Official Review · Reviewer_Gj4g · 2022-10-27

**Confidence:** 4
**Correctness:** 3
**Technical Novelty And Significance:** 2
**Empirical Novelty And Significance:** 3
**Recommendation:** 5

**Clarity, Quality, Novelty And Reproducibility:**

Clarity and Reproducibility: The authors study an interesting and well-motivated problem. The paper is well-written and easy to follow. Sufficient implementation details are provided for reproduction.

Quality: The experiments done in the work indicate the effectiveness of DMAE to some degree. However, additional evaluations on more tasks and datasets may make the method more convincing.

Novelty: The key idea is very straightforward given the problem setting and the proposed approach seems to be an adaptation of MAE in a stricter setting. Hence the novelty is limited.

**Strength And Weaknesses:**

### Strengths

(+) Results on existing tasks are good, showing the effectiveness of the proposed method

(+) The work is well-motivated and the paper is easy to follow.

(+) In addition to the certified accuracy evaluation, the authors include empirical studies to justify that DMAE learns more robust features than MAE.

### Weaknesses

(-) (major) The significance of the work may be limited and it is not well-positioned against existing works (i.e., MAE and Randomized Smoothing). The authors formulate their problem as to study the “robust vision learner” and propose the representation learning (with pre-training & fine-tuning) paradigm to learn robust encoder. The pre-trained encoder should be general-purposed and be evaluated in multiple different downstream tasks, e.g., the robust segmentation on noisy inputs, following the MAE work. However, the current evaluation is only performed on the image classification problem with only 2 datasets. This could limit the significance of the method as a “vision learner“.

(-) (major) The proposed framework that adds noise to the training image and performs the masked reconstruction is straightforward given the desire of learning noise/attack robust encoders. Technically, it seems to be incremental to the MAE work and adapts MAE to the certified robust problem setting. Other techniques used in the paper such as consistency regularization are also from existing work. This may limit the novelty of this work.

(-) (major) The proposed method is not sufficiently justified. Even focusing on the robust classification problem, the experiments may still not be sufficient to show the generalizability of the proposed approach. Current experiments only consider the Gaussian additive noise in both training and evaluation. The certified radius evaluation can cover those cases by selecting a large enough radius, but cannot fully justify or compare the model behavior under certain noise/attack types. In real scenarios, the noise/attack might be more diverse and complicated, e.g., Poisson, impulse, combined, or even dedicated to attacking certain models. It would be better to include more evaluation on some of those cases.

(-) (minor) Does the robust fine-tuning (with regularizations) also be applied to baseline methods so that the comparison is fair? If not, it would be better to make comparisons consistently with/without the robust fine-tuning in order to show that the improvement comes from the DMAE pre-training framework.

(-) (minor) To be more self-contained, it is better to include a better formulation of the certified robust classification problem and the evaluation protocol with the certified radius.

(-) (minor) I also suggest the author discuss the self-supervised (blind-spot) image denoising paper [1, 2] since they are relevant and the frameworks are actually similar. The only difference is that the denoising approaches take the final images as outcomes and this work takes the representation or pre-trained encoder as outcomes. In particular, when considering the masked&noise reconstruction framework and the robust fine-tuning term (KL divergence between two outputs), the framework is very close in its looking to the objective in [2]. It would be better to discuss any connections and highlight differences between those works.

[1] Batson et al. Noise2Self: Blind Denoising by Self-Supervision. ICML 2019.

[2] Xie et al. Noise2Same: Optimizing A Self-Supervised Bound for Image Denoising. NeurIPS 2020.


**Summary Of The Paper:**

In this paper, the authors study the provably/certifiable robust classification problem. Specifically, the authors propose a self-supervised framework called Denoising Masked Auto-encoders to learn robust representations (or to pre-train encoders) by reconstructing images from noisy and masked inputs. The authors further fine-tune the model with the consistency regularization technique to achieve optimal robust classification for the model.

**Summary Of The Review:**

The paper studies an interesting problem and is overall easy to follow. Experiments done in the work can indicate the effectiveness of DMAE to some degree. However, there are some concerns about the novelty, significance, and insufficient evaluation of the work. I believe the paper would be a good one if the authors can show its capability in more tasks with noisy input (that would be a lot of empirical contribution), but the current form of this paper may not be good enough.

---

> ### Author Response · Authors · 2022-11-16
> **To Reviewer Gj4g**
>
> [Regarding the scope of the paper]
>
> We would like to thank the reviewer for pointing out the issue. We have realized that the title confused the readers much. We clarify the scope and contribution of the work as below.
>
> 1) Our work focuses on designing a self-supervised method to improve robust classification tasks. Toward this goal, we integrate the standard MAE with a denoising task, which naturally fits the randomized smoothing certification.
>
> 2) The resulting pre-trained model, DMAE, achieves much better performance with a much smaller model size compared to the SoTA method [Carlini,2022]
>
> To make all the above clear, we change the title to "Denoising Masked Autoencoders Helps Certified Robust Classifications"
>
> **A1:** Thanks for pointing it out. Our former expression of "vision learner" may cause confusion. Following the common practice in the community, we only focus on the classification task to analyze the adversarial robustness. To avoid misunderstanding, we've modified the title to "Denoising Masked Autoencoder Helps Robust Classification" in the revision.
>
> **Q2:** Evaluation on other types of noise/attack.
>
> **A2:** Indeed, the type of adversarial attacks can be more diverse and complicated in real scenarios. But current works in the literature typically focus on the performance under a specific type of attack, otherwise the problem will not be well-formulated. Our work as well as any other method that is based on randomized smoothing only consider $\ell_2$-radius attack. Correspondingly our pre-training task only focus on the Gaussian noise used by the Gaussian smoothed classifier.
>
> **Q3:** Does the robust fine-tuning (with regularizations) also be applied to baseline methods so that the comparison is fair?
>
> **A3:**
> Please note that all baseline methods apply robust fine-tuning, i.e. Gaussian augmentation for classification. Compared with consistency regularization (Jongheon Jeong and Jinwoo Shin), we merely introduce an extra denoising task of pretraining for learning robust features.
>
> **Q4:** Better formulation of the evaluation protocol of the certified radius.
>
> **A4:** We've presented a precise description of the evaluation protocol of the certified radius in the appendix. Please refer to the revision for details.
>
> **Q5:** Reference to two self-supervised (blind-spot) image-denoising papers.
>
> **A5:** Thanks for pointing it out. We've included these two works in reviewing the literature of self-supervised pre-training in vision.

---

> ### Comment · Reviewer_Gj4g · 2022-11-16
> **Thanks for the response**
>
> Thank you for the response and clarifications. I have updated my evaluation to a borderline rating as some of concerns are addressed. While the significance and technical contribution are moderate or limited, the work indeed has some merits on introducing SSL framework into the certified robust classifications problem.

---

### Decision · Program_Chairs · 2023-01-20

**Decision:**

Accept: poster

**Justification For Why Not Higher Score:**

We still have concerns (though minor) about its limited technical contribution and generalization beyond vanilla ViTs. Therefore, we cannot recommend it for a higher score.



**Justification For Why Not Lower Score:**

This paper well demonstrates the potential of MAE in certified robustness, with strong empirical results. Therefore, we believe that this work is worthy of a poster presentation at ICLR.




**Metareview: Summary, Strengths And Weaknesses:**

This paper investigates the potential of Masked Autoencoder (MAE) for certified robustness. Specifically, by training MAE with Gaussian noise corrupted images, this work shows that the MAE's encoder can be used as a base classifier in Gaussian smoothed models and yields significant improvements in certified robustness.

Overall, the reviewer enjoyed reading the paper and appreciated its simplicity and strong empirical results. However, they also had some concerns about: (1) the overclaiming of "robust vision learner", as only the image classification task is studied here; (2) the need for more ablations to justify the effectiveness and efficiency of the proposed method; (3) the comparisons are somewhat unfair; (4) the technical novelty over MAE may be limited; and (5) its applicability beyond vanilla ViTs is unclear. The rebuttal well addressed the concerns (1)-(3). Concern (4)-(5) was discussed in the AC-reviewer meeting and concluded to be minor. As a result, we recommend accepting this paper.

In the final version, the authors are strongly encouraged to elaborate on the possibility of generalizing the proposed method beyond vanilla ViT (either as a future plan or a discussion about current limitations), which can help future readers better understand this paper. In addition, the title and abstract in the OpenReview should be updated to be consistent with the pdf file.

**Note From Pc:**

if the above contains the word "oral" or "spotlight" please see: "oral" presentation means -> notable-top-5% and "spotlight" means -> notable-top-25%. As stated in our emails, we are disassociating presentation type from AC recommendations

**Summary Of Ac-Reviewer Meeting:**

In the meeting, we all agreed that this paper provides a strong and parameter-efficient method for certified robustness. This is an important contribution that would be of interest to the general ICLR audience.

We also reached a consensus that the limited technical contribution over MAE is a minor concern, because the main purpose of this paper is to demonstrate the potential of MAE in certified robustness, which is well justified.

Regarding the concern about extending the proposed method beyond vanilla ViTs, we believe that it is slightly beyond the scope of this paper as this is a general limitation of MAE rather than a limitation specific to this work. An additional discussion of this limitation in the final version is sufficient.

As a result, we all agree to accept this paper.